# Correlation between Serum 25-Hydroxyvitamin D Level and Depression among Korean Women with Secondary Amenorrhea: A Cross-Sectional Observational Study

**DOI:** 10.3390/nu14142835

**Published:** 2022-07-10

**Authors:** Gyung-Mee Kim, Gyun-Ho Jeon

**Affiliations:** 1Department of Psychiatry, Haeundae Paik Hospital, Inje University School of Medicine, Busan 48108, Korea; gmkim2017@gmail.com; 2Department of Obstetrics and Gynecology, Haeundae Paik Hospital, Inje University School of Medicine, Busan 48108, Korea

**Keywords:** vitamin D deficiency, anti-Müllerian hormone, depression, secondary amenorrhea

## Abstract

Vitamin D deficiency is considered a major public health problem worldwide and has been reported as having an association with depression. However, studies on the association between vitamin D deficiency and depressive symptoms in secondary amenorrhea (SA) patients are still scarce. This study examined the relationship between serum 25-hydroxyvitamin D (25(OH)D) levels and depressive symptoms among Korean women with SA. In this cross-sectional observational study, 78 patients with SA were initially recruited. Clinical and biochemical parameters, including serum 25(OH)D level, were measured. Data from 63 SA patients who met the study inclusion criteria and completed psychiatric assessments were finally analyzed. We analyzed their association with depression using a hierarchical regression model. The average serum 25(OH)D level was 34.40 ± 24.02 ng/mL, and 41.3% of the women with SA were vitamin D-deficient (<20 ng/mL). The total score of the Korean version of the Hamilton Depression Rating Scale (K-HDRS) was negatively related to serum 25(OH)D levels, free testosterone, and serum anti-Müllerian hormone (AMH) after adjusting for age and BMI (*r* = −0.450, *p* < 0.001; *r* = −0.258, *p* = 0.045; and *r* = −0.339, *p* = 0.006, respectively). Serum 25(OH)D levels and AMH levels were the most powerful predictors of depressive severity when using the K-HDRS in SA patients (β = −0.39, *p* < 0.005; β = −0.42, *p* < 0.005, respectively). This study showed that low serum 25(OH)D levels were associated with the severity of depressive symptoms in SA patients. This observation suggests that the evaluation of vitamin D deficiency for the risk of depression may be necessary in patients with SA.

## 1. Introduction

Vitamin D is a fat-soluble vitamin associated with calcium metabolism [1] and bone structure [2], and vitamin D deficiency has been considered a major public health problem [3]. According to previous studies, vitamin D is not only essential for bone health, but also plays an important role in blood pressure, glucose control, and immune function, and is even related to cancer, autoimmune diseases, obesity, and depression [4,5,6]. 

In the results of large-scale epidemiological studies and systematic literature reviews, a cross-sectional association between vitamin D levels and depression has been reported [7]. In a large cohort study, 25-hydroxy vitamin D (25(OH)D) levels were related to depressive symptoms in the elderly aged 65 years and older who had severe vitamin D deficiency [8]. In a recent UK cohort study, Ronaldson et al. reported that vitamin D deficiency may be increasing the risk of depression in middle-aged adults [9]. In addition, vitamin D modulates serotonin synthesis [10] and influences immune responses that trigger mood swings by activating stress responses [11]. In this respect, it is thought that vitamin D may be clinically associated with depression. 

Vitamin D is also one of the important factors in the biosynthesis of female hormones such as estrogen [12], as well as in the regulation of anti-Müllerian hormone (AMH) expression in the granulosa cells of hens [13]. Considering these effects of vitamin D on ovarian folliculogenesis and steroidogenesis, vitamin D may be associated with women with secondary amenorrhea who have reproductive steroid imbalances. 

Studies have reported a high prevalence of depression among women with secondary amenorrhea such as polycystic ovarian syndrome (PCOS) or primary ovarian insufficiency (POI). For example, Cooney et al. reported an increased prevalence of depressive symptoms in PCOS [14]. Allshouse et al. reported an increased risk of depression in POI patients [15]. In our previous study, we found that depression in PCOS patients was highly associated with reproductive hormones such as AMH and prolactin [16]. In these respects, vitamin D, which is related to both reproductive hormones and depression, may directly or indirectly affect depression in women with SA. However, studies on the association between vitamin D and depressive symptoms in secondary amenorrhea patients are still scarce.

We hypothesized that low vitamin D levels in women with secondary amenorrhea are likely to be associated with depression. In this study, we compared the differences in clinical characteristics and depressive symptoms according to vitamin D levels in patients with secondary amenorrhea. We also examined which biological factors, including serum vitamin D levels, are related to depressive symptoms in women with secondary amenorrhea.

## 2. Materials and Methods

### 2.1. Participants and Study Design

This study was conducted on a total of 78 patients who were diagnosed with secondary amenorrhea among women who visited a single university hospital in Busan, South Korea, with amenorrhea from March 2018 to February 2019. Secondary amenorrhea was diagnosed by one gynecologic endocrinologist and it was defined as the absence of menstruation for more than three times the period of the previous menstrual cycle, or no menstruation for more than 6 months [17]. For all participants, we obtained their written informed consent, and after informed consent was obtained, further assessments were performed according to the study protocol.

The exclusion criteria in this study were as follows: (1) women who have taken oral contraceptives, vitamin D supplements, thyroid hormones, or antipsychotics within the past 6 months; (2) women who have had ovarian surgery, radiation therapy, or chemotherapy due to gynecological disorders; (3) women who have been diagnosed with a general medical condition, such as diabetes or hyperprolactinemia; (4) women who have severe psychiatric disorders, such as schizophrenia, schizoaffective disorder, psychotic depression, or bipolar disorders with psychotic symptoms; and (5) women unable to complete the questionnaire due to severe cognitive problems, such as neurological disorders, including status epilepticus, or mental retardation and autism spectrum disorder. In consequence, among the 78 SA patients, 6 women due to medications, 2 women with medical or gynecological disorders, 2 women with severe psychiatric disorders, and 2 women with severe cognitive problems were excluded. We also excluded 3 women who declined to participate in this study. The flow chart of this study is presented in Figure 1.

We obtained patients’ sociodemographic information regarding age, parity, menstrual history, current and past medical history, and current and past medications. We also measured the patients’ height, weight, waist circumference, and blood pressure. Body mass index (BMI) was calculated as weight (kg) divided the square of height (m^2^).

To determine the cause of SA, basal gonadotropin hormone levels were measured in all SA subjects, including serum luteinizing hormone (LH), follicle-stimulating hormone (FSH), estradiol, free testosterone, prolactin, and anti-Müllerian hormone (AMH) levels. Each participant’s venous sample was drawn into a serum separation tube (SST) and serum FSH, LH, and estradiol were measured using an Elecsys FSH electrochemiluminescence immunoassay kit with the Cobas e 801 immunoassay device (Roche Diagnostics GmbH, Mannheim, Germany) and presented in mIU/mL. The total imprecision coefficient of variance for FSH, LH, and estradiol was 3.1, 2.1, and 1.7% at a concentration level of 48.8, 51.4 mIU/mL, and 405 pg/m, respectively. AMH levels were measured from separated serum (3000 rpm for 10 min centrifuged) by the Electrochemiluminescent immunoassay (ECLIA) method with an Elecsys AMH kit on a Cobas e 601 immunoassay analyzer (Roche Diagnostics GmbH, Mannheim, Germany) and presented in ng/mL. The total imprecision coefficient of variance was 3.5% at a concentration level of 0.042 ng/mL and 3.4% at 0.20 ng/mL.

For the assessment of a patient’s metabolic status, the serum levels of total cholesterol and fasting glucose were measured. We also assessed their thyroid function tests, including serum-free thyroxine 4 and thyroid-stimulating hormone (TSH) levels. 

To assess the serum level of vitamin D, we measured serum 25-hydroxyvitamin D (25(OH)D) by the Chemiluminescence immunoassay (CLIA) method with a LIAISON 25OH Vitamin D total assay (DiaSorin Inc., Stillwater, Minnesota, USA) on a LIAISON XL (DiaSorin Deutshland DmbH, Dietzenbach, Germany) and presented the results in ng/mL. According to the Endocrine Society clinical practice guidelines, we defined vitamin D deficiency as serum 25(OH)D levels < 20 ng/mL [18].

For psychiatric evaluation, we diagnosed psychiatric disorders of patients including schizophrenia, schizoaffective disorder, psychotic depression, and bipolar disorders with psychotic features using the Diagnostic and Statistical Manual of Mental Disorders, Fifth Edition (DSM-5). To evaluate depressive symptoms, the Korean version of the Center for Epidemiologic Studies Depression Scale (CES-D) was used. The CES-D contains a total of 20 items to assess a patient’s reported depressive symptoms during the past week. We also assessed patients’ depressive symptoms over the past week using the Korean version of the Hamilton Depression Rating Scale (K-HDRS) by a trained psychiatrist. The K-HDRS consists of a 17-item scale, and a higher total score of the K-HDRS means severe depressive symptoms.

Finally, data from 63 patients with SA who completed all of the above assessments were analyzed in the final analysis. The study protocol was approved by the Institutional Review Board of Inje University Haeundae Paik Hospital (No. 2017-01-018).

### 2.2. Statistical Analysis

The data are presented as frequencies with percentages for categorical variables and means ± standard deviations (SDs) for continuous variables. We compared biochemical and depressive parameters according to the serum 25(OH)D levels. Depending on the characteristics of the data, the comparison of continuous variables between groups was conducted using the independent *t*-test and Mann–Whitney U test, and the comparison of categorical variables was achieved though the χ^2^ test with Fisher’s exact test. To check whether its distribution was normal, we used the Shapiro–Wilk test. The partial Spearman’s rank correlation analysis was used to assess the depressive symptoms and biochemical characteristics adjusted for age and BMI. The bivariate normal distribution of each pair of variables was confirmed using Mardia’s test. Hierarchical multiple regression analyses were performed using hormonal and metabolic parameters including vitamin D levels as predictors of depression. All statistics were analyzed using SPSS version 25.0 (SPSS Inc., Chicago, IL, USA) and R4.1.2, and the statistical significance was considered at *p* < 0.05.

## 3. Results

Among the 63 patients with secondary amenorrhea, 40 (63.5%) were diagnosed with polycystic ovarian syndrome, followed by 14 patients (22.2%) with unexplained chronic anovulation, and the remaining 9 (14.3%) were identified as having primary ovarian insufficiency. In this study, the mean age of SA patients was 26.1 ± 8.0 years and their average serum 25(OH)D level was 34.40 ± 24.02 ng/mL. The mean total CES-D score was 17.1 ± 11.6 points, and the total K-HDRS score was 6.4 ± 5.7 points.

### 3.1. Vitamin D Deficiency and Depressive Symptoms

Among the 63 patients included in the study, 26 (41.3%) had vitamin D deficiency (serum 25(OH)D levels < 20 ng/mL), and their mean age was 22.5 ± 5.3 years. The patients in the vitamin D deficiency groups were younger than the patients with normal vitamin D levels (25(OH)D ≥ 20 ng/mL) (Mann–Whitney U = 305.00, *p* = 0.014). There was no statistically significant difference in the depression level and clinical characteristics between the vitamin D deficiency group and the normal vitamin D level group. The mean AMH level was higher in the vitamin D deficiency group than in the normal vitamin D level group, but there was no statistically significant difference in AMH levels between the two groups. The demographic, clinical, and psychological characteristics between the groups are presented in Table 1.

### 3.2. Correlations between Depression and Biochemical Variables

After controlling for age and BMI, the total CES-D score was shown to be negatively related to serum AMH levels and free testosterone levels (*r* = −0.373, *p* = 0.003; and *r* = −0.254, *p* = 0.043, respectively). Although the total CES-D score was also negatively related to serum 25(OH)D levels, there was no statistical significance (*r* = −0.127, *p* = 0.330). The total K-HDRS score showed a negative correlation with serum AMH levels, serum 25(OH)D levels, and free testosterone levels (*r* = −0.450, *p* < 0.001; *r* = −0.258, *p* = 0.045; and *r* = −0.339, *p* = 0.006, respectively). These results are presented in Table 2 and Figure 2.

### 3.3. Predictors of Depression in Women with Secondary Amenorrhea

For the hierarchical multiple regression analyses, age and BMI were entered into the first block of the model. The serum-free testosterone levels were entered into the second block of the model. The serum AMH levels and serum 25(OH)D levels were simultaneously entered as predictors into the third block of the regression model. There was no statistically significant predictor in the first step. In the second step, the results indicated that the serum-free testosterone levels predicted depression after controlling for age and BMI (*R*^2^ = 0.122, ∆*R*^2^ =0.100, F (3, 56) = 2.597, *p* = 0.015). In the final step, we found that the serum AMH levels and serum 25(OH)D levels were the most powerful predictors of depression (β = −0.42 and β = −0.39, respectively). The results are summarized in Table 3. The result of the multivariate regression analysis was not different from that of the hierarchical regression analysis (Appendix A).

## 4. Discussion

The purpose of this study was to examine whether serum 25(OH)D levels were related to depression as well as to identify which biological factors, including serum 25(OH)D levels, were predictors of depression in patients with SA. In this cross-sectional study, we found that depression was negatively related to serum 25(OH)D levels after adjusting for age and BMI. In particular, we found that low serum levels of 25(OH)D and low serum AMH levels were associated with the severity of depression in patients with SA. 

Vitamin D, which is a group of fat-soluble secosteroids, has a significant role in brain function. For example, vitamin D has been shown to have a neuroprotective effect through regulating neurotrophic factors and immunoregulation [19]. Many studies have reported significant results on the association between decreased vitamin D levels and mood disorders [20]. Especially in the elderly, 25(OH)D (calcitriol), which is the active form of vitamin D, is associated with both depressive moods and cognitive decline [21]. Atteritano et al. also reported that the serum levels of 25(OH)D are significantly lower in postmenopausal women with major depressive disorder compared with normal healthy women [22]. 

However, in the case of middle-aged or young women, the results of the study were controversial. For example, it has been reported that decreased vitamin D levels were related to oligomenorrhea and amenorrhea in young women [23,24], and increased depression in these group has been well known [25]. On the other hand, in middle-aged women with the first episode of major depressive disorder, the 25-hydroxyvitamin D levels were not different to normal controls [26]. Additionally, in the results of a recent study that analyzed data from the Korea National Health and Nutrition Examination Survey 2014 of 1736 Korean people, aged from 19 to 76, serum vitamin D levels were only shown to be inversely related to depressive symptoms in men [27]. 

Neither of the studies reported an association between vitamin D deficiency and depression in young women with secondary amenorrhea. This is the first study to investigate the relationship between serum 25(OH)D levels and depression in women with secondary amenorrhea. 

One thing to note when interpreting the results of this study is that geographic differences in vitamin D status should be taken into account to determine whether the association between serum 25(OH)D levels and depression presented in this study is applicable in a global context. In fact, it is well known that the prevalence of major depressive disorder and seasonal affective disorder has been relatively high in countries with a northern latitude, where there is an increased risk for vitamin D deficiency. In contrast, decreased vitamin D levels were also observed in southern countries such as Brazil, India, and Australia, as well as countries in Asia and the Middle East, but not all of those regions have a high prevalence of depressive disorders [28]. However, research reports on this topic in various regions are still insufficient; therefore, based on the results of this study in Korea, additional studies in various regions and further studies on multinational backgrounds will be needed in the future. A recent study by Schaad et al. showed that there was more vitamin D deficiency at higher latitudes than at lower latitudes, and that the prevalence of depression was related to vitamin D deficiency [29]. Because vitamin D synthesis and metabolism associated with depression are influenced by various factors, such as sociodemographic, geographic, genetic, and ethnic factors [30,31,32], a study to investigate the relationship between vitamin D and depression in terms of geographic differences would be complicated. Nevertheless, further well-controlled studies that take these factors into account should be conducted to explain the geographical link between depression and vitamin D.

A possible mechanism by which vitamin D affects depression is that vitamin D deficiency is associated with abnormalities in neurotransmitters [33], including dopamine, serotonin, and norepinephrine, which are the most well-known mechanisms of depression. Moreover, vitamin D receptors exist in the brain regions, such as the prefrontal cortex, hippocampus, and cerebellum, that are impaired in neuropsychiatric disorders [34,35]. In addition, there is a study that shows that vitamin D response elements are found in the promoter regions of serotonin genes [36]. According to animal studies, there were motor impairments, increased grooming behavior, and anxiety-like behaviors among vitamin D-receptor-deficient mice [37,38]. Therefore, there is a possibility that depressive symptoms may occur due to the functional deterioration of neurotransmitters due to vitamin D deficiency. Although there have been some studies that support a biological link between vitamin D and depression [39,40,41,42], additional studies are necessary for further understanding of the underlying pathogenesis of the association between vitamin D and depression.

Furthermore, we found that AMH is another important factor in predicting depression in SA patients. This result is consistent with our previous study. We reported a negative correlation between serum AMH concentration and depression severity in our previous study among women with PCOS [16]. Additionally, similar results of the negative association between AMH and depressive symptoms were also reported in women with salpingo-oophorectomy, and even among young and nulliparous women [43,44]. AMH, expressed by the granulosa cells in ovarian follicles, is usually measured for evaluating ovarian reserve function in patients with secondary amenorrhea [45]. According to previous animal studies, AMH may have neuroprotective and neuroregenerative action by increasing the activity of GnRH on the hypothalamic–pituitary–gonadal axis, which is an important mechanism in depression [46]. However, the direction of the relationship between depression and serum AMH levels and that underlying its pathogenesis have not yet been clarified. Therefore, further studies are needed to confirm the relationship between AMH and depression and elucidate the underlying biological mechanism connecting AMH and depression.

Interestingly, in this study, the mean AMH levels were higher in the vitamin D-deficient (serum 25(OH)D < 20 ng/mL) group than in the normal vitamin D level (serum 25(OH)D ≥ 20 ng/mL) group, though there was no statistically significant difference in the AMH levels between the two groups. The results of previous studies on the association between vitamin D deficiency and serum AMH levels have been inconsistent. For example, Drakopoulos et al. found no relationship between vitamin D and AMH in 283 women with infertility [47]. Meanwhile, Merhi et al. reported that serum vitamin D levels were positively related to serum AMH levels [48]. Because the sample size is small, the data of serum AMH levels are not normally distributed, and a relatively large proportion (63.5%) of participants in our study had PCOS, further studies with a larger population are needed to confirm the relationship between serum 25(OH)D level and AMH.

This study has several limitations, and caution is required in the interpretation of its results. Firstly, we should include normal healthy women with normal menstruation as controls. Due to the lack of comparison with normal controls, the generalizability of the results in this study is limited. Secondly, there is a risk of selection bias due to the relatively small number of study subjects and the difference in the number of subjects for each subgroup by cause of secondary amenorrhea. The study population consists of a heterogeneous group that may increase type II errors. Thirdly, this study was conducted as a study design of the prospective observational cohort in nature, so we should not interpret these results in terms of causality. Further studies with a larger population and compared with a normally healthy control group should be conducted for more definite conclusions.

Despite these limitations, this study is meaningful as it is the first piece of research, to our knowledge, to investigate the relationship between serum 25(OH)D levels and depression among young women with secondary amenorrhea. Based on the results of this study, it is worth considering that patients with secondary amenorrhea who have lower vitamin D levels are more likely to have more severe depressive symptoms. Additionally, in this case, it can be expected that, even if there is no osteoporosis, vitamin D supplementation would be meaningful to help decrease the risk of depression in patients with SA who have low serum levels of vitamin D. According to a recent systematic review and meta-analysis study, vitamin D supplementation lasting two months with a dosage of ≤4000 IU/day has an effect on reducing negative emotions, including depression [49]. However, the results included in this study are highly heterogeneous, such that conclusions have not yet been reached. Therefore, a study with more subjects and considering various symptoms or disease groups is needed.

## 5. Conclusions

In summary, we found that serum 25(OH)D levels and AMH levels negatively related to the severity of depression in SA patients. The results of this study suggest that patients with secondary amenorrhea who have lower vitamin D levels and lower AMH levels may be more likely to have more severe depressive symptoms. Future studies could examine whether depression is modifiable by vitamin D supplementation without pharmacotherapy, including hormone therapy and antidepressants, in SA patients.

## Figures and Tables

**Figure 1 nutrients-14-02835-f001:**
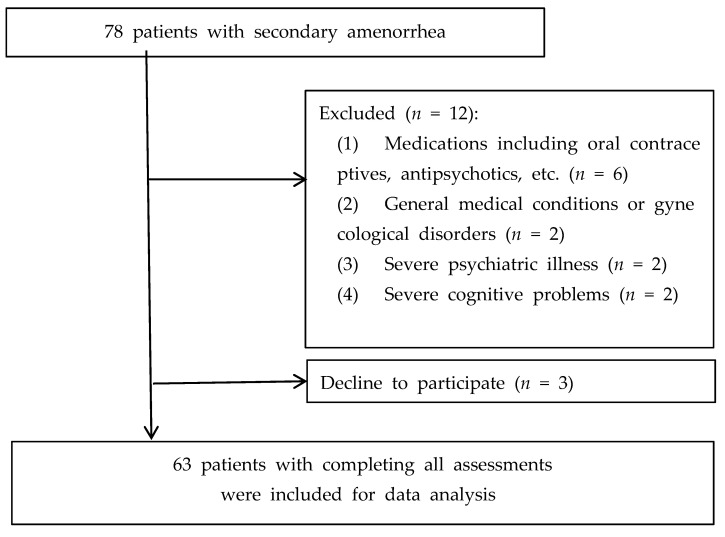
Flow chart of the study population.

**Figure 2 nutrients-14-02835-f002:**
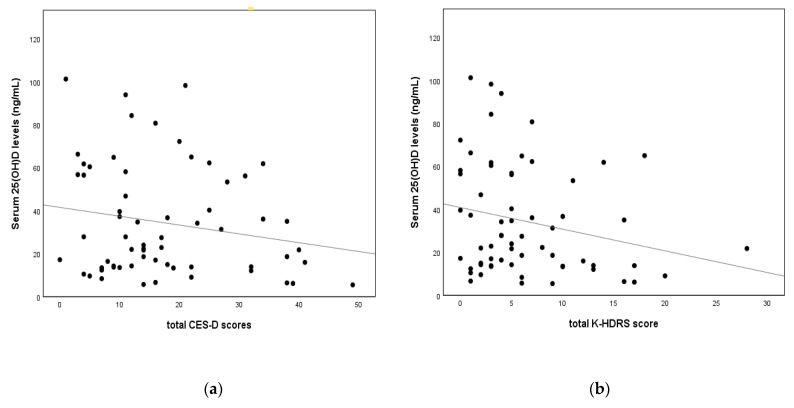
Correlation between depression severity and serum 25-hydroxyvitamin D levels after adjusting for age and body mass index. (**a**) Serum 25(OH)D levels and total CES-D score (*r* = −0.127, *p* = 0.330); (**b**) serum 25(OH)D levels and total K-HDRS score (*r* = −0.256, *p* = 0.045). Abbreviations: CES-D—Center for Epidemiological Studies–Depression Rating Scale, K-HDRS—Korean version of the Hamilton Depression Rating Scale, 25(OH)D—25-hydroxyvitamin D.

**Table 1 nutrients-14-02835-t001:** Demographic and clinical characteristics according to serum 25-hydroxyvitamin D levels among 63 women with secondary amenorrhea.

Variable		Serum 25-Hydroxyvitamin D(25(OH)D) Levels		*p*-Value/χ^2^
		25(OH)D < 20 ng/mL (*n* = 26)	25(OH)D ≥ 20 ng/mL (*n* = 37)	
Demographic factors				
Age (years)		22.54 ± 5.33	27.81 ± 8.65	0.014 ^2^
BMI (kg/m^2^)		22.85 ± 5.17	22.55 ± 4.39	0.823 ^1^
Menarche (years)		13.42 ± 1.06	13.53 ± 1.33	0.909 ^2^
Parity	0	25 (48.1%)	27 (51.9%)	0.069 ^3^
	1	1 (14.3%)	6 (85.7%)	
	2	0 (0%)	4 (100%)	
Hormonal status				
Estradiol (pg/mL)		63.18 ± 68.30 (*n* = 25)	67.77 ± 63.28 (*n* = 36)	0.866 ^2^
Free testosterone (pg/mL)		2.19 ± 0.83	1.82 ± 0.79	0.077 ^1^
Prolactin (ng/mL)		17.84 ± 11.54	16.79 ± 15.61	0.135 ^2^
AMH (ng/mL)		10.86 ± 8.94 (*n* = 25)	7.24 ± 5.62 (*n* = 35)	0.130 ^2^
LH (U/L)		18.96 ± 21.04 (*n* = 25)	13.81 ± 12.42 (*n* = 36)	0.127 ^2^
FSH (U/L)		7.78 ± 7.70	7.67 ± 9.37	0.917 ^2^
TSH (μmol/L)		2.05 ± 0.97 (*n* = 24)	2.16 ± 1.32 (*n* = 36)	0.815 ^2^
fT4 (ng/dL)		1.25 ± 0.22 (*n* = 24)	1.24 ± 0.19 (*n* = 36)	0.988 ^1^
Metabolic parameters				
Total cholesterol (mg/dL)		184.04 ± 38.84 (*n* = 26)	192.62 ± 30.42 (*n* = 36)	0.334 ^1^
Fasting glucose (mg/dL)		99.36 ± 33.69 (*n* = 25)	90.86 ± 9.29 (*n* = 36)	0.116 ^2^
Psychiatric assessments				
Total CES-D score		18.77 ± 13.56	16.54 ± 10.37	0.753 ^2^
Total K-HDRS score		7.35 ± 5.89	5.81 ± 5.69	0.317 ^2^

^1^: *p*-values were derived from an independent *t*-test. ^2^: *p*-values were derived from the Mann–Whitney U test. ^3^: Fisher’s exact test. Abbreviations: 25(OH)D—serum 25-hydroxyvitamin D, BMI—body mass index, AMH—anti-Müllerian hormone, LH—luteinizing hormone, FSH—follicle-stimulating hormone, TSH—thyroid-stimulating hormone, fT4—free thyroxine 4, CES-D—Center for Epidemiological Studies–Depression Rating Scale, K-HDRS—Korean version of the Hamilton Depression Rating Scale.

**Table 2 nutrients-14-02835-t002:** Correlation between depression scores and biochemical variables after adjusting for age and body mass index.

Variable	CES-D		K-HDRS	
	*r*	*p*-Value	*r*	*p*-Value
25(OH)D(ng/mL)	−0.127	0.330	−0.258	0.045 *
Estradiol (pg/mL)	−0.077	0.552	0.049	0.705
Free testosterone (pg/mL)	−0.254	0.043 *	−0.339	0.006 *
Prolactin (ng/mL)	0.026	0.836	0.174	0.168
AMH (ng/mL)	−0.373	0.003 **	−0.450	<0.001 **
LH (U/L)	−0.104	0.420	−0.216	0.092
FSH (U/L)	−0.017	0.897	−0.128	0.315
TSH (μmol/L)	0.002	0.986	0.136	0.297
fT4 (ng/dL)	0.187	0.150	0.083	0.525
Total cholesterol (mg/dL)	0.005	0.971	0.083	0.518
Fasting glucose (mg/dL)	0.061	0.639	0.127	0.325

*r*: Spearman’s correlation coefficient, *: *p* < 0.05, **: *p* < 0.005. Abbreviations: CES-D—Center for Epidemiological Studies–Depression Rating Scale, K-HDRS—Korean version of the Hamilton Depression Rating Scale, AMH—anti-Müllerian hormone, LH—luteinizing hormone, FSH—follicle-stimulating hormone, 25(OH)D—25-hydroxyvitamin D, TSH—thyroid-stimulating hormone, fT4—free thyroxin 4.

**Table 3 nutrients-14-02835-t003:** Hierarchical multiple regression analysis.

	CES-D				K-HDRS			
Predictor	B	SE	β	*R* ^2^	B	SE	β	*R* ^2^
Step 1				0.01				0.02
Age	−0.15	0.21	−0.10		0.04	0.10	0.06	
BMI	0.18	0.38	0.06		−0.21	0.18	−0.15	
Step 2				0.08 *				0.12 *
Age	−0.21	0.20	−0.14		0.00	0.10	0.01	
BMI	0.22	0.37	0.08		−0.19	0.18	−0.14	
Free testosterone	−3.76	1.84	−0.27 *		−2.24	0.89	−0.32 *	
Step 3				0.25 **				0.21 **
Age	−0.19	0.21	−0.06		0.11	0.10	0.15	
BMI	0.17	0.34	0.06		−0.22	0.16	−0.16	
Free testosterone	−0.56	2.02	−0.04		−0.87	0.94	−0.12	
AMH	−0.72	0.23	0.46 **		−0.33	0.11	−0.42 **	
25(OH)D (ng/mL)	−0.12	0.06	−0.26		−0.09	0.03	−0.39 **	

*: *p* < 0.05, **: *p* < 0.005. Abbreviations: AMH—anti-Müllerian hormone, BMI—body mass index, CES-D—Center for Epidemiological Studies–Depression Rating Scale, K-HDRS—Korean version of the Hamilton Depression Rating Scale, 25(OH)D—25-hydroxyvitamin D.

## Data Availability

Some or all of the datasets generated and/or analyzed during the current study are not publicly available, but are available from the corresponding author upon reasonable request.

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
