# Peer review of "Correlation between Serum 25-Hydroxyvitamin D Level and Depression among Korean Women with Secondary Amenorrhea: A Cross-Sectional Observational Study"

_nutrients, 2022, doi:10.3390/nu14142835_

Round 1
Reviewer 1 Report
The authors evaluated associations between serum 25(OH)D and depression scores in women specifically with secondary amenorrhea and then conducted regression analyses to determine 25(OH)D was a significant predictor of depression, among other variables, again in women with secondary amenorrhea. To assess depressive symptoms, the authors used two different depression scales. AMH and free T tended to be increased in the vitamin D deficient group vs normal Vit D concentrations, however no differences were noted between groups across depression scores. The authors report strong negative associations between depression scores and AMH, and between vitamin D and one of the two depression scores. The authors appropriately acknowledge their findings are heterogeneous in the conclusion but lack a clear message in the body of the manuscript.
Abstract:
The sample size in the abstract (N=78) does not match the sample size in the final analysis (N=63). This should be clarified.
Introduction:
The introduction did not sufficiently motivate the study; the authors poorly established biological plausibility between vitamin D and secondary amenorrhea, and secondary amenorrhea (broadly speaking) and depression.
Methods:
The authors were not clear which subtypes of secondary amenorrhea were present among the recruited cohort, which makes interpretation of findings particularly challenging. The authors did not define how secondary amenorrhea was diagnosed by the provider.
It is still unclear how secondary amenorrhea was defined and which sub-classifications of secondary amenorrhea comprised the group.
Regarding sample size, the authors should list reasons for exclusion to account for the loss of 15 participants in the final analysis.
The authors should clarify exclusion of mental health conditions when evaluating associations between mental health and mental health status among women with secondary amenorrhea (lines 67-69).
Assay details for reproductive hormones are not provided and should be included.
Why was IGF-1 measured in this study? Please clarify.
Can the authors justify why a control population was not included? This would be an important inclusion to address a critical gap in this study: whether vitamin D is associated with depressive systems in general.
Discussion:
The instructions for the discussion are included in the first paragraph of the manuscript (lines 162-165).
The authors did not address the body of literature that conflicts with their findings, that PCOS is associated with both increased AMH and depression symptoms.
Line 206-207: AMH did not predict depression in SA patients. AMH was a predictor of depressive symptoms. Please clarify.
Lines 223-227: I believe the authors overextend their findings by claiming that this study supports vitamin D supplementation for depression in secondary amenorrhea. I think the authors need to directly address the notion that correlation is not causation and what that means for this study.
Could the authors consider their findings in the broader global context? Are associations between vitamin D and depression in women generally present in northern countries verses southern countries? The external validity of this finding, given the geographical relevance of vitamin D status, should be addressed.
Overall, the rationale, clinical and physiological implications, and consideration of false equivalence need to be addressed. This is overall an interesting research question but substantial edits should be made to the manuscript to clarify the contribution to science. A critical concern of this manuscript is its lack of inclusion of a control group.
Author Response
Response to Reviewer 1 Comments
We appreciate the reviewers who provided valuable insight regarding our manuscript. The comments were very useful for improving the quality of our manuscript. We revised our manuscript according to the reviewer’s suggestions, and the revised manuscript with tracked changes and yellow-colored highlight has been uploaded. The responses to the reviewer’s comments are as follows.
Point 1: # Abstract: The sample size in the abstract (N=78) does not match the sample size in the final analysis (N=63). This should be clarified.
Response 1: We appreciate your comments and we apologize for the confusion. In this study, seventy-eight patients with secondary amenorrhea (SA) were initially recruited. Among those, we used data from sixty-three patients, who completed all assessments and did not be excluded by the exclusion criteria and who refused to participate in this study, for the final analysis. We described this briefly in the abstract as follows:
[Abstract]
“…. In this cross-sectional observational study, seventy-eight patients with SA were initially recruited.…. Data from 63 SA patients, who met the study inclusion criteria and completed psychiatric assessments were finally analyzed. ….”
Point 2: Introduction:
The introduction did not sufficiently motivate the study; the authors poorly established biological plausibility between vitamin D and secondary amenorrhea, and secondary amenorrhea (broadly speaking) and depression.
Response 2: We appreciate your comments. We edited the introduction to clarify our research motivation with the biological relationship between vitamin D and secondary amenorrhea, and between SA and depression as follows:
[In the Introduction]
“ Vitamin D is also one of the important factors ….. Considering of these effects of vitamin D on ovarian folliculogenesis and steroidogenesis, vitamin D may be associated with secondary amenorrhea women who have reproductive steroid imbalances.
Studies have reported a high prevalence of depression among women with secondary amenorrhea such as polycystic ovarian syndrome (PCOS) or primary ovarian insufficiency (POI). For example, Cooney et al. reported that increased prevalence of depressive symptoms in PCOS [11]. Allshouse et al. reported an increased risk of depression in POI patients [12]. In our previous study, we found that depression in PCOS patients was highly associated with reproductive hormones such as AMH and prolactin [13]. In these respects, vitamin D, which is related to both reproductive hormones and depression, may directly or indirectly affect depression in women with SA. However, studies on the association ….”.
Point 3: Methods:
The authors were not clear which subtypes of secondary amenorrhea were present among the recruited cohort, which makes interpretation of findings particularly challenging. The authors did not define how secondary amenorrhea was diagnosed by the provider.
It is still unclear how secondary amenorrhea was defined and which sub-classifications of secondary amenorrhea comprised the group.
Response 3: We appreciate your comments and we added the following contents in the Method and Results section, respectively as the reviewer pointed as follows:
[In the Materials and Methods]
“…. Secondary amenorrhea was diagnosed by one gynecologic endocrinologist and it was defined as the absence of menstruation for more than 3 times the period of the previous menstrual cycle or no menstruation for more than 6 months……."
[In the Results]
“Among the 63 patients with secondary amenorrhea, 40 (63.5%) were diagnosed with the polycystic ovarian syndrome, followed by 14 patients (22.2%) with unexplained chronic anovulation, and the remaining 9 (14.3%) were identified as primary ovarian insufficiency….."
Point 4: Regarding sample size, the authors should list reasons for exclusion to account for the loss of 15 participants in the final analysis.
Response 4: We appreciate your comments. Of those 78 SA patients, six women who have taken medication including oral contraceptives, vitamin D supplements, or antipsychotics within the past 6 months, two women who have been diagnosed with a general medical condition such as diabetes or hyperprolactinemia, or with gynecological problems such as ovarian surgery, chemotherapy, or radiation therapy, two women who have severe psychiatric disorders such as schizoaffective disorder and bipolar disorder with psychotic symptoms, and two women who have severe cognitive problems such as neurological disorders including epilepsy, or mental retardation were excluded. We also excluded three women who declined to participate in this study. we added the following contents in the Method and Results section, respectively as the reviewer pointed out. We added the flow chart of this study, which explains the specific reason of excluded 15 participants in summary[Figure 1].
[In the Materials and Methods]
“The exclusion criteria in this study were as follows: 1) women who have taken oral contraceptives, vitamin D supplements, thyroid hormone, or antipsychotics within the past 6 months, 2) women who have had ovarian surgery, chemotherapy, or radiation therapy due to gynecological disorders, 3) women who have been diagnosed with a general medical condition such as diabetes or hyperprolactinemia, 4) women who have severe psychiatric disorders such as schizophrenia, schizoaffective disorder, psychotic depression, or bipolar disorders with psychotic symptoms, and 5) women unable to complete the questionnaire due to severe cognitive problems such as neurological disorders including status epilepticus, or mental retardation and autism spectrum disorder. In consequence, among those 78 SA patients, six women due to medications, two women with medical or gynecological disorders, two women with severe psychiatric disorders, and two women with severe cognitive problems were excluded. We also excluded three women who declined to participate in this study. The flow chart of this study is presented in Figure 1.”
Point 5: The authors should clarify exclusion of mental health conditions when evaluating associations between mental health and mental health status among women with secondary amenorrhea (lines 67-69).
Response 5: We appreciate your comments. In this study, we excluded women who have taken antipsychotics within the past 6 months, or who have had severe psychiatric disorders such as schizophrenia, schizoaffective disorder, psychotic depression, and bipolar disorders with psychotic symptoms. Women with mild depression or with anxiety disorders were not excluded from this study. We used the Diagnostic and Statistical Manual of Mental Disorders, fifth edition (DSM-5) for psychiatric diagnosis. If the exclusion criteria were not met, all participants were assessed for depressive symptoms using both subjective measurement (the Korean version of the Center for Epidemiologic Studies Depression Scale) and objective measurement by trained a psychiatrist (the Korean version of Hamilton Depression Rating Scale). We added these in the materials and methods section as follows:
[In the Materials and Methods]
“For psychiatric evaluation, we diagnosed psychiatric disorders of patients including schizophrenia, schizoaffective disorder, psychotic depression, and bipolar disorders with psychotic symptoms using the Diagnostic and Statistical Manual of Mental Disorders, fifth edition (DSM-5). To evaluate depressive symptoms,….”
Point 6: Assay details for reproductive hormones are not provided and should be included.
Response 6: We appreciate your comments. We added the following contents in the Method section as the reviewer suggested.
[In the Materials and Methods]
“ .. Each participant’s venous sample was drawn into an SST and serum FSH, LH, and estradiol were measured using an Elecsys FSH electrochemiluminescence immunoassay kit (Roche Diagnostics GmbH, Mannheim, Germany) with an immunoassay device (Cobas e 801) and presented in mIU/mL. The total imprecision coefficient of variance for FSH, LH, and estraiol were 3.1, 2.1, and 1.7% at a concentration level of 48.8, 51.4, mIU/mL, and 405 pg/m, respectively. AMH level was measured from separated serum (3,000 rpm for 10 minutes centrifuged) by ECLIA method with Elecsys AMH kit (Roche Diagnostics GmbH) on a Cobas e 601 immunoassay analyzer and presented in ng/mL. The total imprecision coefficient of variance was 3.5% at a concentration level of 0.042 ng/mL and 3.4% at 0.20 ng/mL. “
Point 7: Why was IGF-1 measured in this study? Please clarify.
Response 7: We appreciate the reviewer’s valuable comments about IGF-1 measurement. We assessed the IGF-1 levels of patients with polycystic ovary syndrome (PCOS). In PCOS, it is known that the increased insulin concentration due to insulin resistance inhibits the hepatic secretion of IGF-binding protein-1 and increases the IGF-1 concentration. Because the patients with PCOS which account for the majority of patients with secondary amenorrhea were included in this study (63.5%), we compared IGF-1 levels between the vitamin D deficiency group and the normal vitamin D level group. However, we decided to delete the presentation of this result in the revised documents because IGF-1 levels were not to be an important factor for evaluating all SA patients’ status in this study, and there were no statistically significant differences either.
Point 8: Can the authors justify why a control population was not included? This would be an important inclusion to address a critical gap in this study: whether vitamin D is associated with depressive systems in general.
Response 8: We appreciate your comments. We agree with the reviewer's opinion that we should include the general women without any related infertility symptoms as controls. However, in reality, it was difficult to recruit healthy women who volunteered to measure laboratory tests including various hormonal levels, metabolic parameters, and vitamin D levels for the study alone as a control group. Rather, we focused that secondary amenorrhea patients have a higher risk of depression than the normal healthy women. As mentioned in the introduction, under the background that vitamin D can affect both female reproduction and depression, the result that vitamin D could be a significant factor affecting depression scores in patients with secondary amenorrhea can be considered meaningful. Nevertheless, we agree that the main limitation of generalizing these results to the general population is that normal controls were not included.
We added the limitations of our study in the discussion section as follows:
[in the Discussion section]
"…. This study has several limitations and needs to be cautious in the interpretation of the results. Firstly, we should include the normal healthy women with normal menstruation as control…..Further studies with a larger population and compared with a normally healthy- control group should be conducted for more definite conclusions….”
Point 9: Discussion: The instructions for the discussion are included in the first paragraph of the manuscript (lines 162-165).
Response 9: It is thankful for your review of our manuscript and we apologize that we couldn’t more carefully double-check before submission. We deleted the instructions for discussion in this revised version.
Point 10: The authors did not address the body of literature that conflicts with their findings, that PCOS is associated with both increased AMH and depression symptoms.
Response 10: We appreciate your comments about the relationship between AMH and depression in PCOS patients. We agree with the reviewer’s comment that PCOS is associated with abnormally increased AMH and that the risk of depression is high in PCOS patients. Though AMH level is increased in PCOS women, we found a negative correlation between serum AMH concentration and depression severity in our previous study among women with PCOS (Kim et al. 2021). And similar results of a negative association between AMH and depressive symptoms were also reported in women with salpingo-oophorectomy, and even among young and nulliparous women. In our study, though 63.5% of patients in this study have PCOS, 36.3% of non-PCOS women who have decreased AMH levels were also included. This heterogeneity may influence the direction of the relationship between AMH and depression in this study. We added the following contents in the discussion section and limitation of our study, respectively as the reviewer pointed as follows:
[in the Discussion section]
“….We reported the negative correlation between serum AMH concentration and depression severity in our previous study among women with PCOS (GM Kim et al. 2021). And similar results of the negative association between AMH and depressive symptoms were also reported in women with salpingo-oophorectomy, and even among young and nulliparous women. ….”
"…. Study population consists of a heterogeneous group may increase of type II error. ….”
Point 11: Line 206-207: AMH did not predict depression in SA patients. AMH was a predictor of depressive symptoms. Please clarify.
Response 11: We appreciate your comments about the relationship between AMH and depression in SA patients. We agree that the relationship between AMH and depression is not yet confirmed. However, according to recent animal studies (reference No. 37 and No. 38), AMH may have neuroprotective and neuroregenerative action by increasing the activity of GnRH on the hypothalamic-pituitary-gonadal axis. In our previous study, we found that depressive scores correlated negatively with serum AMH levels in SA patients (Jeon HG and Kim GM, 2018). According to a recent prospective cohort study by Golenbock SW, et al.(2020), they also found the negative association between depression and AMH levels. In this study, we found that AMH is another important factor in predicting depression in SA patients. This result may be one of the evidences supporting the relationship between AMH and depression. We agree with the reviewer’s comment that AMH cannot be definitively predictive of depression, and further studies are needed. We added the following contents in the discussion section, respectively as the reviewer pointed as follows:
[in the Discussion section]
“Furthermore, we found that AMH is …. This result is consistent with our previous study. We reported the negative correlation between serum AMH concentration and depression severity in our previous study among women with PCOS [13]. And similar results of the negative association between AMH and depressive symptoms were also reported in women with salpingo-oophorectomy, and even among young and nulliparous women [31, 32]….. Therefore, further studies are needed to confirm the relationship between AMH and depression and to elucidate the underlying biological mechanism relating AMH and depression.“
Point 12: Lines 223-227: I believe the authors overextend their findings by claiming that this study supports vitamin D supplementation for depression in secondary amenorrhea. I think the authors need to directly address the notion that correlation is not causation and what that means for this study.
Response 12: It is very thankful for the reviewer’s concern that we might not be overemphasizing our results. This study is a cross-sectional observational study, so our results cannot be interpreted as causal inferences. Though there was a recent systematic review and meta-analysis study about the effectiveness of vitamin D supplements for decreasing depressive symptoms, our study results didn’t mean the effectiveness of vitamin D supplementation for depression in SA patients. Therefore, further study should be needed for supporting vitamin D supplementation for depression in SA. We added the limitation of our study with this aspect at the end of the Discussion section as follows:
[in the Discussion section]
"….. Thirdly, this study was conducted as a study design of prospective observational cohort in nature, so we should not interpret these results in terms of causality..….”
Point 13: Could the authors consider their findings in the broader global context? Are associations between vitamin D and depression in women generally present in northern countries verses southern countries? The external validity of this finding, given the geographical relevance of vitamin D status, should be addressed.
Response 13: We appreciate for reviewer’s valuable comments about taking into account geographic differences in vitamin D status when explaining the associations between vitamin D and depression. It is well-known that the prevalence of major depressive disorder and of seasonal affective disorders are relatively high in Northern latitude countries where increased risk for vitamin D deficiency. In contrast, vitamin deficiency was also observed in southern counties as well as Asia or Middle Eastern countries. However, the prevalence of depression in these countries varies from country to country. As the reviewers pointed out, geographic differences in vitamin D status should be taken into account to determine whether the association between vitamin D and depression presented in this study is applicable in a global context. However, research reports on this topic in various regions are still insufficient, so based on the results of this study in Korea, additional studies in various regions and further studies on multinational backgrounds will be needed. We added the following contents in the Discussion section as the reviewer suggested.
[in the Discussion section]
“One thing to note when interpreting the results of this study is geographic differences in vitamin D status should be taken into account to determine whether the association between vitamin D and depression presented in this study is applicable in a global context. In fact, it is well-known that the prevalence of major depressive disorder and of seasonal affective disorders have been relatively high in northern latitude countries where increased risk for vitamin D deficiency. In contrast, decreased vitamin D levels were also observed in southern countries such as Brazil, India, and Australia, as well as countries in Asia or Middle Eastern, but not all of these regions have a high prevalence of depressive disorder [24]. However, research reports on this topic in various regions are still insufficient, so based on the results of this study in Korea, additional studies in various regions and further studies on multinational backgrounds will be needed in the future. A recent study by Schaad KA et al. presented that the vitamin D deficiency in the higher latitudes was more than that in the lower latitudes and the prevalence of depression was related to vitamin D deficiency [25]. Because vitamin D synthesis and metabolism associated with depression is influenced by various factors such as socio-demographic, geographic, genetic, and ethnic factors, a study to investigate the relationship between vitamin D and depression in terms of geographic differences would be complicated. Nevertheless, the further well-controlled studies that take these factors into account should be needed to explain the geographical link between depression and vitamin D.”
Point 14: Overall, the rationale, clinical and physiological implications, and consideration of false equivalence need to be addressed. This is overall an interesting research question but substantial edits should be made to the manuscript to clarify the contribution to science. A critical concern of this manuscript is its lack of inclusion of a control group.
Response 14: We appreciate your comments. We agree with the reviewer's comments, and we added the limitations and caution of interpretation of this study in the Discussion section as below.
[in the Discussion section]
" This study has several limitations and needs to be cautious in the interpretation of the results. Firstly, we should include the normal healthy women with normal menstruation as control. Secondly, there is a risk of selection bias due to the relatively small number of study subjects and the difference in the number of subjects for each subgroup by cause of secondary amenorrhea. The study population consists of a heterogeneous group that may increase type II error. Thirdly, this study was conducted as a study design of the prospective observational cohort in nature, so we should not interpret these results in terms of causality. Further studies with a larger population and compared with a normally healthy- control group should be conducted for more definite conclusions.”
For consideration of publication in Nutrients as an original article, we are pleased to revise the attached manuscript “Correlation between serum 25-Hydroxyvitamin D level and Depression among Korean women with secondary amenorrhea: A cross-sectional observational study”.
We hope you find this report acceptable for publication in Nutrients.
Sincerely yours
Gyun-Ho Jeon. M.D. Ph.D.
Reviewer 2 Report
This cross-sectional investigation examined the link between vitamin D deficiency and secondary amenorrhea among a patient population in Busan, South Korea. Previous research had linked vitamin D deficiency with depression in older adults. However, since vitamin D influences the synthesis and regulation of female hormones, there was speculation that vitamin D would improve mood disorders in premenopausal women with secondary amenorrhea. This study collected sociodemographic information, multiple biomarker data in blood, serum vitamin D data, and depressive symptom scores on 63 women diagnosed with secondary amenorrhea. The paper requires careful editing for grammar as there are widespread grammar violations.
Specific concerns are itemized below.
The paper title is poorly written; the title should be a declarative statement reflecting the results of the study.
The abstract contains data that do not appear to reflect what is stated in the text (lines 19-22). This needs to be corrected to accurately reflect the data therein.
Line 33: ‘preventing’ is a strong word – rather vitamin D lowers risk for these chronic conditions.
Methods: Participant flow is not clearly presented. 78 women were diagnosed with SA – however, 63 participants represent the study data. State the causes for attrition – how many did not meet the study criteria? How many declined participation? Was written consent obtained? These items needs to be clearly stated in the paper.
The authors state that they will determine the cause of SA in their participants (line 82); however, these data are not presented and discussed. Please include this information.
Details regarding blood analyses are missing in the methods section (second paragraph of methods).
Tables 1 and 2: multivariate analyses should be used for data analysis due to the many measures and the relations among these measurements. Are data normally distributed and meet the assumptions of parametric testing?
Results: AMH was elevated in the vitamin D deficient group (p=0.059). Is this unexpected? Please discuss these findings in the discussion.
Delete the 1st paragraph on the discussion.
Please add a study limitation section to the discussion.
Author Response
Response to Reviewer 2 Comments
We appreciate the reviewers who provided valuable insight regarding our manuscript. The comments were very useful for improving the quality of our manuscript. We revised our manuscript according to the reviewer’s suggestions, and the revised manuscript with tracked changes and the yellow-colored highlights has been uploaded. The responses to the reviewer’s comments are as follows.
Point 1: The paper title is poorly written; the title should be a declarative statement reflecting the results of the study.
Response 1: We appreciate the reviewer’s valuable comments and as the reviewer’s recommendation, we would like to change the research title of our study as follows:
[Title]
Correlation Between Serum 25-Hydroxyvitamin D Level And Depression Among Korean Women With Secondary Amenorrhea: A Cross-sectional Observational Study
Point 2: The abstract contains data that do not appear to reflect what is stated in the text (lines 19-22). This needs to be corrected to accurately reflect the data therein.
Response 2: It is very thankful for your careful review of our manuscript and we apologize that we couldn’t more carefully double-check any wrong expression of the results before submission. The abstract contains data presented in Table 2 and Table 3. We edited the abstract as follows:
[Abstract]
“…The total score of the Korean version of Hamilton Depression Rating Scale (K-HDRS) was negatively related to serum 25(OH)D level, free testosterone, and Serum anti-Müllerian hormone (AMH) after adjusting age and BMI (r=-0.294, p=0.001; r=-0.428, p=0.001; r=-0.303, p=0.015; respectively). Serum AMH level and 25(OH)D level were the most powerful predictors of depressive severity using the K-HDRS in SA patients (β = -0.42, p<0.005; β = -0.39, p<0.005; respectively)….”
Point 3: ‘preventing’ is a strong word – rather vitamin D lowers risk for these chronic conditions.
Response 3: We appreciate the reviewer’s comments and we reworded the document as follows:
[Introduction]
“ …..and even lowers risk for cancer, autoimmune diseases, obesity, and depression”
Point 4: Methods: Participant flow is not clearly presented. 78 women were diagnosed with SA – however, 63 participants represent the study data. State the causes for attrition – how many did not meet the study criteria? How many declined participation? Was written consent obtained? These items need to be clearly stated in the paper.
Response 4: We appreciate your valuable comments and we apologize for the confusion. Initially, we recruited seventy-eight women with SA after informed consent was obtained. In the initial document, we describe the informed consent statement at ‘Back Matter”. In this revised document, we added the informed consent statement in the Materials and Methods section as follows:
[in the Materials and Methods]
“ … For all participants, we provided their written informed consent, and after informed consent was obtained, the further assessments were performed according to the study protocol.”
In this study, seventy-eight patients with SA were initially recruited. Among those, six women with their medication, two women with medical or gynecological problems, two women with severe psychiatric illness, and two women with severe cognitive problems were excluded. We also excluded three women who declined to participate in this study. Finally, we analyzed data from sixty-three patients with SA who completed all assessments. We added the flow chart of the study subject, which explains the specific reasons of excluded 15 participants [Figure 1]. We edited the document in the Methods section as follows:
[in the Materials and Methods]
“The exclusion criteria in this study were as follows: 1) women who have taken oral contraceptives, vitamin D supplements, thyroid hormone, or antipsychotics within the past 6 months, 2) women who have had ovarian surgery, radiation therapy, or chemotherapy, due to gynecological disorders, 3) women who have been diagnosed with a general medical condition such as diabetes or hyperprolactinemia, 4) women who have severe psychiatric disorders such as schizophrenia, schizoaffective disorder, psychotic depression, or bipolar disorders with psychotic symptoms, and 5) women unable to complete the questionnaire due to severe cognitive problems such as neurological disorders including status epilepticus, or mental retardation and autism spectrum disorder. In consequence, among those 78 SA patients, six women due to medications, two women with medical or gynecological disorders, two women with severe psychiatric disorders, and two women with severe cognitive problems were excluded. We also excluded three women who declined to participate in this study. The flow chart of this study is presented in Figure 1.”
Point 5: The authors state that they will determine the cause of SA in their participants (line 82); however, these data are not presented and discussed. Please include this information.
Response 5: It is very thankful for your careful review of our manuscript. We added the following contents in the Results section, respectively as the reviewer pointed out.
[in the Result]
"…. Among the 63 patients with secondary amenorrhea, 40 (63.5%) were diagnosed with the polycystic ovarian syndrome, followed by 14 patients (22.2%) with unexplained chronic anovulation, and the remaining 9 (14.3%) were identified as primary ovarian insufficiency…."
Point 6: Details regarding blood analyses are missing in the methods section (second paragraph of methods).
Response 6: We appreciate your comments. We added the following contents in the Materials and Methods section as reviewer’s comments:
[in the Method section]
“To determine the cause of SA, basal gonadotropin hormone levels were measured in all SA subjects, including serum luteinizing hormone (LH), follicle stimulating hormone (FSH), estradiol, free testosterone, prolactin, and anti-Müllerian hormone (AMH) level. Each participant’s venous sample was drawn into a serum separation tube (SST) and serum FSH, LH, and estradiol were measured using an Elecsys FSH electrochemiluminescence immunoassay kit (Roche Diagnostics GmbH, Mannheim, Germany) with an immunoassay device (Cobas e 801) and presented in mIU/mL. The total imprecision coefficient of variance for FSH, LH, and estraiol were 3.1, 2.1, and 1.7% at a concentration level of 48.8, 51.4, mIU/mL, and 405 pg/m, respectively. AMH level was measured from separated serum (3,000 rpm for 10 minutes centrifuged) by ECLIA method with Elecsys AMH kit (Roche Diagnostics GmbH) on a Cobas e 601 immunoassay analyzer and presented in ng/mL. The total imprecision coefficient of variance was 3.5% at a concentration level of 0.042 ng/mL and 3.4% at 0.20 ng/mL.
Point 7: Tables 1 and 2: multivariate analyses should be used for data analysis due to the many measures and the relations among these measurements. Are data normally distributed and meet the assumptions of parametric testing?
Response 7: We appreciate for reviewer’s valuable comments about our statistical methods and we agree with your comments. We re-analyzed our data with the consultation of a statistical specialist for this revised version of our study. We use the Mann-Whitney U test to compare differences between the vitamin D deficiency group and the normal vitamin D group when the variable is not normally distributed. We edited the prescription about statistical analysis in the Materials and Method section and changed the presentation of Table 1 according to the results by revised analysis methods.
We agree with the reviewer's opinion that we should analyze using multivariate regression due to multiple variables. We sought to determine which of several biochemical factors including vitamin D associated with depression in patients with SA. So we used the partial correlation to find the depression and each biochemical variable after controlling for age and BMI. Our purpose of this study was to find out which of these variables was the most associated with depression, so we use the hierarchical multiple regression analyses. When we used the multivariate regression analysis, the results were the same as those in our initial documents using hierarchical multiple regression analyses. AMH and vitamin D levels were related to a total score on the Hamilton Depression Rating Scale. We added the results of multivariate regression analysis as supplementary materials, respectively as the reviewer pointed out.
[in the Materials and Method section]
“… Depending on the characteristics of the data, the comparison of continuous variables between groups was analyzed using independent t-test and Mann-Whitney U test,….”
Point 8: Results: AMH was elevated in the vitamin D deficient group (p=0.059). Is this unexpected? Please discuss these findings in the discussion.
Response 8: It is very thankful for your careful review of our data and results in this study. We think there was an error we didn’t normality test for our data in the previous document. We should have looked into the data more closely with received statistical consultation. We re-analyzed our data with the consultation of a statistical specialist. Differences in clinical and biochemical characteristics between the vitamin D deficiency group and the normal vitamin D level group using independent t-test and Mann-Whitney U test. These results were presented in Table 1. As the result of the Mann-Whitney U test, though the mean AMH level is higher in the vitamin D deficient group than in the normal vitamin D level group, there was no statistically significant difference in AMH level between the two groups (p=0.130). Because of the small sample size of our study, it is impossible to generalize these results. We added this result in the Results section as follows:
[in Result section]
“ …. The mean AMH level is higher in the vitamin D deficiency group than in the normal vitamin D level group but there was no statistically significant difference in AMH level between two groups….”
In addition, previous studies on the relationship between vitamin D and AMH levels have had inconsistent results. For example, Merhi et al. found a positive relationship between serum vitamin D and AMH levels (Merhi ZO, Seifer DB, Weedon J, Adeyemi O, Holman S, Anastos K, et al. Circulating vitamin D correlates with serum antimullerian hormone levels in late-reproductiveaged women: Women’s Interagency HIV Study. Fertil Steril 2012;98:228-34). Meanwhile, a recent cross-sectional study that included 283 infertile women, revealed no significant association between vitamin D and AMH (Drakopoulos P, van de Vijver A, Schutyser V, Milatovic S, Anckaert E, Schiettecatte J, et al. The effect of serum vitamin D levels on ovarian reserve markers: a prospective cross-sectional study. Hum Reprod 2017;32:208-14.). Though vitamin D may influence gonadal function through AMH signaling (Merhi Z, Doswell A, Krebs K, Cipolla M. Vitamin D alters genes involved in follicular development and steroidogenesis in human cumulus granulosa cells. J Clin Endocrinol Metab 2014;99:E1137-45.), further study is needed to confirm the relationship between vitamin D levels and AMH. We added this result in the Discussion section.
[in the Discussion section]
“Interestingly, in this study, the mean AMH level is higher in the vitamin D deficient group than in the normal vitamin D level group, though there was no statistically significant difference in AMH level between the two groups. The results of previous studies about the association between vitamin D deficiency and serum AMH level have been inconsistent. For example, Drakopoulos et al. found no relationship between vitamin D and AMH in 283 women with infertility. Meanwhile, Merhi et al. reported that serum vitamin D level was positively related to serum AMH level. Because the sample size is small and the data of serum AMH levels are not normally distributed in our study, further study with a larger population is needed to confirm the relationship between vitamin D levels and AMH.”
Point 9: Delete the 1st paragraph on the discussion.
Response 9: It is thankful for your review of our manuscript and we apologize that we couldn’t more carefully double-check before submission. We deleted the instructions for discussion in this revised version.
Point 10: Please add a study limitation section to the discussion.
Response 10: We appreciate the reviewer’s comments on our manuscript. We added the study limitation at the end of the Discussion section.
[in the Discussion section]
"This study has several limitations and needs to be cautious in the interpretation of the results. Firstly, we should include the normal healthy women with normal menstruation as control. Secondly, there is a risk of selection bias due to the relatively small number of study subjects and the difference in the number of subjects for each subgroup by cause of secondary amenorrhea. The study population consists of a heterogeneous group that may increase type II error. Thirdly, this study was conducted as a study design of the prospective observational cohort in nature, so we should not interpret these results in terms of causality. Further studies with a larger population and compared with a normally healthy- control group should be conducted for more definite conclusions.”
For consideration of publication in Nutrients as an original article, we are pleased to revise the attached manuscript “The vitamin D status correlates with depression in women with secondary amenorrhea: A cross-sectional observational study”.
We hope you find this report acceptable for publication in Nutrients.
Sincerely yours
Gyun-Ho Jeon. M.D. Ph.D.
Round 2
Reviewer 1 Report
Overall, there have been several improvements to the manuscript and have addressed all comments. However, some concerns persist.
The authors are writing about observational/cross-sectional studies in a causal manner, which is inappropriate. I would strongly urge the authors to review their citations and the literature carefully and rephrase their introduction accordingly. Examples include:
Line 32-33: this sentence is lacking appropriate references.
Line 35-36: "even lowers risk for.." - I would soften this language. The studies cited are reviews and a SR of observational, not interventional studies. Thus, it seems like the studies cited reflect correlation, not causation.
Line 39-40: "resulted in" is inappropriate as this was a survey-based dataset and reflects associations only. (citation 5)
Citation 10 is described in the context of humans but only reflects observations in the hen.
Methods:
Line 70-73: Please clarify and cite diagnostic guidelines for secondary amenorrhea if any were used.
Line 104: ECLIA or ELISA?
Table 1, parity: a question for the authors: does it make more sense to present parity as a percentage within each Vit D cohort? To understand the distribution or parity within each cohort? This is not a criticism, just a thought.
Discussion
You find that lower vit D and lower AMH both are independent predictors of higher depressive symptoms. Line 205-206 you should clarify this inverse predictive relationship, particularly because elevated AMH is associated with PCOS, and PCOS is strongly (positively) associated with depression scores. The authors might also consider their finding that the vit D deficient cohort had elevated AMH might be explained by the type of secondary amenorrhea (i.e., relative proportions of participants with PCOS vs other conditions?).
Several sentences throughout are not properly referenced (i.e., line 247-249, line 256-259, as examples only).
Reviewer 2 Report
The paper is improved. The lack of a control group is the major shortcoming of the research which has been acknowledged by the authors.
Author Response
Response to Notes of Academic Editor #1
We appreciate the comments of the academic editor #1. We revised our manuscript according to the academic editor’s suggestions, and the revised manuscript with tracked changes. The responses to the experts’ comments are as follows.
Point 1: The authors included several revisions to the manuscript to address reviewers' concerns. However, reviewers' concerns that "the authors overextend their findings" do not appear to have been adequately addressed. Lines 213-214, 304-308, and 315-216 might be misleading to some readers, especially given the limitations of the study design and limited generalizability of the study outcomes.
Response 1: We appreciate your comments. We agree with editor #1's concerns that there may be a risk of misinterpretation or of overgeneralization. In response to the editor #1’s concerns, we edited the manuscript as follows.
Lines 213-214: In particular, serum 25(OH)D levels and serum AMH levels were the most powerful predictors of depression among women with SA.
- The purpose of this study….. In particular, we found that low serum levels of 25(OH)D and low serum AMH levels were associated with severity of depression in patients with SA.
Lines 304-308: Based on the results of this study, we should consider that patients with secondary amenorrhea, even in young women, are more likely to have more severe depressive symptoms with greater vitamin D deficiency. And, in this case, it can be expected that, even if there is no osteoporosis, the vitamin D supplement for correcting vitamin D deficiency can help recovery from depression.
- Based on the results of this study, it is worth considering that patients with secondary amenorrhea who have lower vitamin D levels are more likely to have more severe depressive symptoms. Additionally, in this case, it can be expected that, even if there is no osteoporosis, vitamin D supplementation would be meaningful to help decrease the risk of depression in patients with SA who have low serum levels of vitamin D.
Lines 315-316: In summary, we found that serum 25(OH)D levels and AMH levels were the two strongest predictor of depression in SA patients.
- In summary, we found that serum 25(OH)D levels and AMH levels negatively related to the severity of depression in SA patients. The results of this study suggest that patients with secondary amenorrhea who have lower vitamin D levels and lower AMH levels may be more likely to have more severe depressive symptoms.
This study has several limitations and needs to be cautious in the interpretation of the results. Firstly, we should include the normal healthy women with normal menstruation as control. Secondly, there is a risk of selection bias……
- …… Firstly, we should include the normal healthy women with normal menstruation as control. Due to the lack of comparison with normal controls, the generalizability of the results in this study is limited. Secondly, there is a risk of selection bias……
We hope you find our report acceptable for publication in Nutrients.
Sincerely yours
Gyun-Ho Jeon. M.D. Ph.D.
Response to Notes of Academic Editor #3
We appreciate the comments of the academic editor #3. We revised our manuscript according to the academic editor’s suggestions, and the revised manuscript with tracked changes. The responses to the experts’ comments are as follows.
Point 1: Authors should do a throrough grammar checking in the manuscript. For example, in the abstract, the first sentence is grammatically incorrect. It should say: Vitamin D deficiency is considered a major public health problem worldwide. Among other problems an association with depression has been reported. There are other grammatical problems that need to be fixed
Response 1: We appreciate your comments. For English proofreading, we used the MDPI English editing service (Invoice ID:english-46529). Our manuscript has been checked for correct use of grammar and we have edited the manuscript as corrected by the MDPI English editing service. The newly revised manuscript after receiving English proofreading was uploaded.
Point 2:. Authors should also address the comment rasied by the editors before this manuscript can be considered for acceptance.
Response 2: It is thankful for your concerns. For the second revision of our manuscript, we did our best to respond to the opinions of reviwer #2 and academic editors. However, there seems to have been an error on the e-submission homepage. We had not permitted access to the e-submission webpage, so we sent our second revised manuscript, a brief cover letter for editors, and a reply to the reviewer’s comments by email.
For our third revised manuscript, we edited our manuscript to reflect the concerns of academic editor #1 about misinterpretation. And we edited the part of the limitation of our study as the response to the 2nd opinion of reviewer #2. As mentioned above, we also received English proofreading.
We are pleased to be able to upload our third revised manuscript on the e-submission homepage. We hope you find our report acceptable for publication in Nutrients.
Sincerely yours
Gyun-Ho Jeon. M.D. Ph.D.
Response to Reviewer 2 Comments
Point 1: The paper is improved. The lack of a control group is the major shortcoming of the research which has been acknowledged by the authors.
Response 1: We appreciate your careful review of our manuscript and we agree with reviewer 2’s concerns of the limitation (the lack of a control group in our study). In our second revised manuscript, the lack of a control group was described as the first limitation of this study, and the following sentence was added in our third revised manuscript.
- …… Firstly, we should include the normal healthy women with normal menstruation as control. Due to the lack of comparison with normal controls, the generalizability of the results in this study is limited. Secondly, …..……
For consideration of publication in Nutrients as an original article, we are pleased to revise the attached manuscript “The vitamin D status correlates with depression in women with secondary amenorrhea: A cross-sectional observational study”.
We hope you find this report acceptable for publication in Nutrients.
Sincerely yours
Gyun-Ho Jeon. M.D. Ph.D.